# Early and late onset pre-eclampsia and small for gestational age risk in subsequent pregnancies

**Thomas P. Bernardes**[1]*, **Ben W. Mol**[2], **Anita C. J. Ravelli**[3], **Paul van den Berg**[4], **H. Marike Boezen**[1], **Henk Groen**[1]

1 Epidemiology, University Medical Center Groningen, Groningen, Netherlands, 2 Obstetrics and Gynaecology, Monash University, Clayton, Australia, 3 Medical Informatics/Obstetrics and Gynaecology, Amsterdam UMC, Amsterdam, Netherlands, 4 Obstetrics and Gynaecology, University Medical Center Groningen, Groningen, Netherlands

* thpatrick@gmail.com

**Data Availability Statement:** Data used in this study belongs to the Dutch Perinatal Registry (https://www.perined.nl/). Access is subject to approval by the Dutch Perinatal Registry (Perined).

## Abstract

### Background

Pre-eclampsia shares pathophysiology with intrauterine growth restriction.

### Objective

To investigate whether delivery of a small for gestational age (SGA) infant in the 1st pregnancy increases the risk of early and late onset pre-eclampsia in the 2nd pregnancy. Conversely, we investigated whether pre-eclampsia in the 1st pregnancy impacts SGA risk in the 2nd pregnancy.

### Study design

We studied a cohort from the Dutch Perinatal Registry of 265,031 women with 1st and 2nd singleton pregnancies who delivered between 2000 and 2007. We analyzed 2nd pregnancy risks of early and late onset pre-eclampsia—defined by delivery before or after 34 gestational weeks—as well as SGA below the 5th and between the 5th and 10th percentiles risks with multivariable logistic regressions. Interaction terms between 1st pregnancy hypertension, pre-eclampsia, SGA, and delivery before or after 34 gestational weeks were included in the regressions.

### Results

First pregnancy early onset pre-eclampsia increased risk of SGA <5th percentile (OR 2.1, 95% CI 1.7–2.7) in the 2nd pregnancy. Late onset pre-eclampsia increased the SGA <5th percentile marginally (OR 1.1, 95% CI 1.0–1.3). In the absence of 1st pregnancy hypertensive disorder, women who delivered an SGA infant in their 1st pregnancy were at increased risk of 2nd pregnancy late onset pre-eclampsia (SGA <5th: OR 2.05, 95% CI 1.58–2.66; SGA 5–10th: OR 1.39, 95% CI 1.01–1.93). Early onset 2nd pregnancy pre-eclampsia risk was also increased, but this was only statistically significant for women who delivered an SGA

Perined grants access to researchers who meet its criteria for access to confidential data. The authors confirm that they did not receive special access privileges to the data that others would not have. Researchers may contact Perined at Mercatorlaan 1200, 3528 BL Utrecht (phone: 030 - 3690800, email: info@perined.nl).

**Funding:** The author(s) received no specific funding for this work.

**Competing interests:** The authors have declared that no competing interests exist.

infant below the 5th percentile in the 1st pregnancy (SGA <5th: OR 2.44, 95% CI 1.19–5.00; SGA 5–10th: OR 1.69, 95% CI 0.68–4.24;).

## Conclusion

Women with 1st pregnancy early onset pre-eclampsia have increased risk of SGA <5th percentile in the 2nd pregnancy. SGA in the 1st pregnancy increases pre-eclampsia risk in the 2nd pregnancy even in the absence of hypertensive disorders in the 1st pregnancy, although absolute risks remain low. These findings strengthen the evidence base associating intrauterine growth restriction with early onset pre-eclampsia.

## Introduction

Globally, one in twenty pregnancies is complicated by pre-eclampsia.[1] Its occurrence imposes significant morbidity and mortality risks on both mother and fetus, especially in developing countries.[2,3] The severity of adverse outcomes has a strong association with gestational age of onset. Occurrence late in the pregnancy is generally associated with better outcomes, while early onset often leads to unfavorable results.[4–6] Differing pathophysiological processes have been hypothesized to justify the difference in timing. Early onset pre-eclampsia has been associated with poor placentation and dysfunctional spiral artery remodeling. These are uncommonly found in late onset pre-eclampsia, which tends to be milder, and may occur without placental dysfunction.[7,8] Furthermore, evidence of poor placentation is not pathognomonic of pre-eclampsia, as it can also be found in association with pregnancies with no features of pre-eclampsia but which were complicated by fetal growth restriction.[9,10]

Nonetheless, this common pathophysiological feature favors parallel occurrence of pre-eclampsia and intrauterine growth restriction.[11,12] Diagnosis of pre-eclampsia without severe features should currently trigger ultrasonographic investigation of the growth restriction, while evidence of intrauterine growth restriction warrants close observation for the subsequent development of pre-eclampsia.[13,14] Furthermore, it is well established that women with pre-eclampsia in a previous pregnancy have high risk of recurrence.[15–19] Similarly, delivery of a small for gestational age (SGA) infant is associated with a higher risk of intrauterine growth restriction in subsequent pregnancies.[20–23]

The use of low dose aspirin from 12–16 weeks of gestation in women screened as high risk for either pre-eclampsia or intrauterine growth restriction is now recommended by multiple guidelines.[14,24–28] Identification of women who might benefit is usually based on the presence of one or more risk factors. In addition to these, the use of biomarkers such as maternal serum pregnancy-associated plasma protein A and placental growth factor as well as other measurements such as mean arterial pressure, uterine-artery pulsatility index has been included in more complex screening algorithms.[29] Whether such algorithms are cost effective in comparison to other screening mechanisms or even a policy of low dose aspirin in every pregnancy has been put into question.[30]

Presence of some risk factors, such as previous pregnancies affected by pre-eclampsia or intrauterine growth restriction are now considered sufficient to prompt the intervention with low dose aspirin. Evidence supporting other risk factors such as nulliparity, obesity and family history of pre-eclampsia is weaker, and intervention requires the combined presence of two or more factors. Efforts to clarify the relative importance of these risk factors have been limited so far and do not provide enough evidence to further refine treatment decisions.[18]

Given the hypothesized pathophysiological similarities and prevention potential with aspirin, we aimed to evaluate whether delivery of an SGA infant affected the risk of early and late onset pre-eclampsia in a subsequent pregnancy, and conversely, if occurrence of early and late onset pre-eclampsia in the previous pregnancy increases SGA risk.

## Methods

We conducted this study on population-based prospective cohort data that covers approximately 96% of all deliveries in the Netherlands. These data were obtained from Perined, a national registry that contains validated linked data of three different Dutch registries: the midwifery registry (LVR1), the obstetrics registry (LVR2), and the neonatology registry (LNR). It consists of information on pregnancies, deliveries and admissions up to 28 days after birth. No individual informed consent was obtained as only anonymous registry data was used in this study. The Dutch Perinatal Registry approved the use of the data in this study (approval no. 12.56).

There is no unique maternal identifier in Perined data that would allow us to identify siblings and outcomes in subsequent pregnancies as data is registered at the child's level. For this reason, a linkage procedure was performed on all available deliveries from Jan 1, 2000 to Dec, 28 2007. The procedure was based on the following variables: birth date of mother, birth date of previous child, and postal code of mother. The resulting linked cohort dataset contained information on the first and second deliveries of women. Further details on the 2000–2007 longitudinal linkage procedure can be found elsewhere. [31]

SGA was defined following Dutch reference charts by partiy, gender and ethnicity.[32] In this study SGA for infants with birthweights below the 10th and below the 5th percentiles were used. Combined presence of hypertension (either maximum diastolic blood pressure $\geq$90 mmHg or documented hypertension by the care provider) and proteinuria ($\geq$300 mg in 24 h) were the criteria for pre-eclampsia following definitions at the time of data collection. Otherwise documented pre-eclampsia was also included. Hypertension diagnosed before pregnancy or new onset hypertension before 20 weeks of pregnancy were considered as chronic hypertension following Dutch guidelines and as recorded by the care providers, either a midwife or obstetrician. Early and late onset pre-eclampsia were characterized by delivery before 34 weeks and from 34 weeks onwards, respectively, in women with pre-eclampsia.

We studied the occurrence of early and late onset pre-eclampsia in the second pregnancy and its association with 10th and 5th percentiles SGA infants and additional potential risk factors present in the first pregnancy. We also studied early and late onset pre-eclampsia occurrence in the first pregnancy as risk factors for delivery of an SGA infant in the subsequent pregnancy. To adjust for potential confounders the following 1st pregnancy clinical and demographic characteristics were included in the regressions: maternal age (years), gestational age at delivery (weeks and before/after 34 weeks of gestational age), non-Caucasian ethnicity (yes or no), low socioeconomic status (yes or no), any cause hypertension (yes or no), pre-eclampsia (yes or no), chronic hypertension (yes or no), pre-gestational diabetes (yes or no), gestational diabetes (yes or no), placental abruption (yes or no), HELLP syndrome (yes or no), assisted reproduction (yes or no), spontaneous labor (yes or no), stillbirth (yes or no), neonatal mortality (yes or no), congenital abnormalities (yes or no). None of the analyzed variables contained missing values. Mann-Whitney U and Chi-square tests were used for continuous and categorical data, respectively. All variables were first evaluated with univariable logistic regressions.

In the multivariable logistic regressions for the occurrence of early and late onset pre-eclampsia in the 2nd pregnancy, we assessed potential interaction effects between hypertension,

pre-eclampsia, SGA below the $5^{th}$ percentile and between the $5^{th}$ and $10^{th}$ percentiles. In the multivariable logistic regressions of delivery of SGA below the $5^{th}$ percentile and between the $5^{th}$ and $10^{th}$ percentiles we evaluated potential interaction effects between delivery before completion of 34 weeks of gestation, hypertension and pre-eclampsia in the $1^{st}$ pregnancy. Interaction effects were evaluated following the same methodology we used in previously published work on this cohort that implements an alternative coding scheme initially proposed by Rothman and that was further developed by Hosmer & Lemeshow.[19,33] In general terms, the interaction between two risk factors (A and B) is assessed through a single four-level variable (-A-B, +A-B, -A+B, +A+B), with no loss of degrees of freedom. Point estimates for each combination and associated confidence intervals are easier to interpret than with traditional interaction analysis. The record linkage procedure was performed using the R statistical software environment (version 2.13.1; R Foundation for Statistical Computing, Vienna, Austria). Statistical analyses were performed with IBM SPSS Statistics software (version 25.0.0; IBM Corporation).

## Results

Data was available for 265,031 (97%) first and second singleton pregnancies from the longitudinal linked cohort.[31] There were 6375 (2.4%) women who presented with pre-eclampsia in the first pregnancy, of which 853 (0.32% of 265,031) had early onset. In the second pregnancy, 2362 (0.9%) women presented with pre-eclampsia, of whom 201 (0.07% of 265,031) delivered before the $34^{th}$ week. The prevalence of $10^{th}$ and $5^{th}$ percentiles SGA closely followed the appropriate percentiles: 9.7% and 5.1% in the first pregnancy, and 9% and 4.5% in the second. Further descriptive and analytical results are shown divided in four sections. The first section shows descriptive data by pre-eclampsia occurrence in the first pregnancy; these serve as reference for the results in the second section: analysis of the impact of pre-eclampsia occurrence in the first pregnancy on the risk of SGA delivery in the subsequent pregnancy. Similarly, the third section shows descriptive data by SGA delivery in the first pregnancy, followed by the fourth section with analysis of the effects of SGA delivery in the first pregnancy in the risk of pre-eclampsia in the subsequent pregnancy.

### Descriptive characteristics by time of occurrence of pre-eclampsia in the first pregnancy

Table 1 presents baseline demographics, comorbidities, pregnancy characteristics and neonatal outcomes according to gestational age at delivery and pre-eclampsia occurrence among women in their first pregnancy. Median maternal age was similar in the four groups, with the median age for women who presented with early onset pre-eclampsia being one year less than the other three. Median gestational age at delivery was lower by construction in the delivery before 34 weeks group and early onset pre-eclampsia group. Late onset pre-eclampsia was also associated with a lower median gestational age at delivery. Non-caucasian women were overrepresented in both groups with delivery before 34 weeks. Low socioeconomic status was less common in the group that delivered before 34 weeks and did not develop pre-eclampsia. Higher rates of SGA in the $5^{th}$ to $10^{th}$ percentile range were found in late onset pre-eclampsia as well as delivery before 34 weeks with or without pre-eclampsia. The same occurred with SGA below the $5^{th}$ percentile, and the biggest difference found was in the late onset pre-eclampsia group. In the absense of pre-eclampsia, hypertension was more common in the group that delivered before 34 weeks. Chronic hypertension was more frequent in the pre-eclampsia groups, especially in early onset. Placental abruption was more common before 34 weeks of gestation, but was also observed in late onset pre-eclampsia. HELLP syndrome was

**Table 1. Baseline characteristics at 1st pregnancy delivery by preeclampsia occurrence.**

| | Delivery ≥ 34 weeks of gestation | | | | Delivery < 34 weeks of gestation | | | |
|---|---|---|---|---|---|---|---|---|
| | No pre-eclampsia (n = 253,518) | | Pre-eclampsia (n = 5,519) | | No pre-eclampsia (n = 5,143) | | Pre-eclampsia (n = 851) | |
| Maternal age, years† | 29 | (26–31) | 29 | (26–31) | 29 | (26–31) | 28 | (25–31) |
| Gestational age at delivery, weeks† | 40 | (38–41) | 38 | (37–39) | 31 | (28–33) | 31 | (29–32) |
| Non-caucasian, n (%) | 32,290 | 12.7% | 657 | 11.9% | 754 | 14.7% | 117 | 13.7% |
| Low socioeconomic status, n (%) | 64,896 | 25.6% | 1,345 | 24.4% | 1,178 | 22.9% | 219 | 25.7% |
| SGA 5–10th percentile, n (%) | 11,083 | 4.4% | 477 | 8.6% | 324 | 6.3% | 107 | 12.6% |
| SGA <5th percentile, n(%) | 12,403 | 4.9% | 679 | 12.3% | 445 | 8.7% | 66 | 7.8% |
| Hypertension | 38,490 | 15.2% | 5,519 | 100.0% | 984 | 19.1% | 851 | 100.0% |
| Chronic hypertension, n (%) | 2,171 | 0.9% | 346 | 6.3% | 76 | 1.5% | 86 | 10.1% |
| Chronic diabetes, n (%) | 2,448 | 1.0% | 123 | 2.2% | 69 | 1.3% | 12 | 1.4% |
| Gestational diabetes, n (%) | 1,528 | 0.6% | 63 | 1.1% | 23 | 0.4% | 5 | 0.6% |
| Placental abruption, n (%) | 120 | 0.05% | 20 | 0.4% | 78 | 1.5% | 18 | 2.1% |
| HELLP syndrome, n (%) | 522 | 0.2% | 280 | 5.1% | 94 | 1.8% | 127 | 14.9% |
| Assisted reproduction, n (%) | 53,824 | 21.2% | 1,611 | 29.2% | 1,322 | 25.7% | 220 | 25.9% |
| Spontaneous labor, n (%) | 176,412 | 69.6% | 3,420 | 62.0% | 4,529 | 88.1% | 810 | 95.2% |
| Stillbirth, n (%) | 1,284 | 0.5% | 25 | 0.5%* | 1,009 | 19.6% | 77 | 9.0% |
| Neonatal mortality, n (%) | 599 | 0.2% | 12 | 0.2%* | 584 | 11.4% | 44 | 5.2% |
| Congenital abnormalities, n (%) | 5,684 | 2.2% | 175 | 3.2% | 621 | 12.1% | 67 | 7.9% |

SGA: small for gestational age. HELLP syndrome: hemolysis, elevated liver enzymes, and low platelet count syndrome

† Given as median and interquartile range

* Not statistically different compared with delivery at 34 or more weeks of gestation with a 95% confidence interval.

particularly present in pre-eclampsia, especially in cases with early onset. Assisted reproduction rates were higher in the three comparison groups, with the highest rate found in the late onset pre-eclampsia group. Spontaneous labor, stillbirth, neonatal mortality and congenital abnormalities were more commonly observed in case of delivery before 34 weeks of gestation.

## Risk of SGA in the second pregnancy by gestational age at delivery, and by hypertension or pre-eclampsia occurrence in the 1st pregnancy

Results of the multivariable regressions for delivery of an SGA infant in the 2nd pregnancy with birthweights between the 5th and 10th percentiles and below the 5th percentile, presented by gestational age at delivery and the interaction with hypertension or pre-eclampsia occurrence in the 1st pregnancy are found in Fig 1. The risks of SGA in the 2nd pregnancy associated with delivery of moderately or severely SGA infant in the 1st pregnancy are also presented in Fig 1. Delivery before the 34th week in the 1st pregnancy was associated with increased risk of both SGA categories in the 2nd pregnancy. If the delivery in the 1st pregnancy occurred after the 34th week, hypertension in the 1st pregnancy did not substantially raise these risks, and neither did pre-eclampsia. Women who developed hypertension in their 1st pregnancy and delivered before the 34th week were at increased risk of SGA in the 2nd pregnancy, although the effect size for SGA in the 5–10th percentiles was similar to those that did not present hypertension but delivered before completion of 34 weeks of gestation. On the other hand, the combination of these two factors resulted in additional risk of SGA below the 5th percentile in the subsequent pregnancy, when compared to women who delivered before 34 weeks but did not

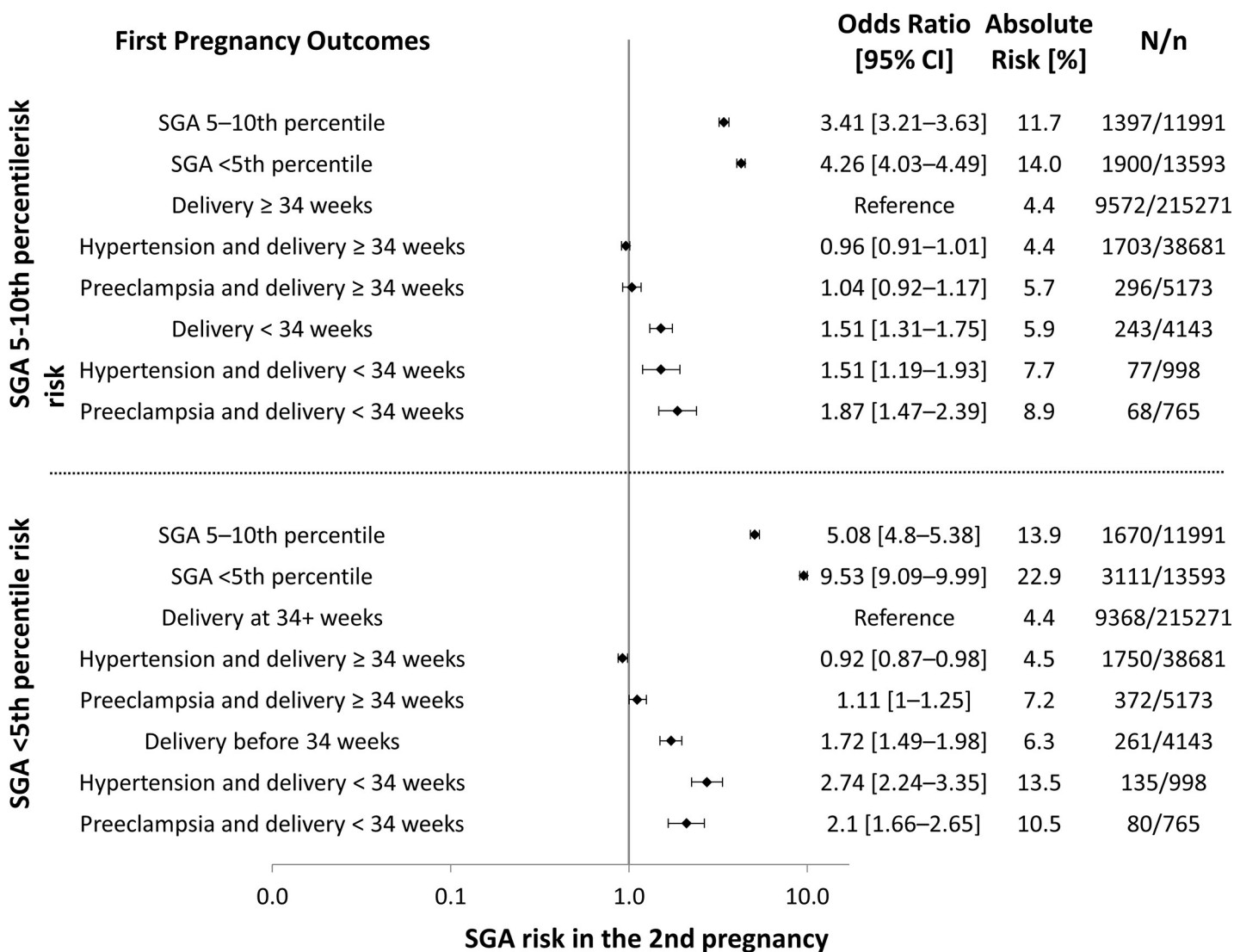

**Fig 1. SGA risk in the 2nd pregnancy by gestational age at delivery, hypertension and pre-eclampsia in the 1st pregnancy.** Second pregnancy odds ratios and absolute risk of SGA between the 5[th] and 10[th] percentile (top) and below the 5[th] percentile (bottom) by groups according to gestational age at delivery, occurrence of hypertension and pre-eclampsia in the first pregnancy. SGA: small for gestational age. CI: confidence interval.

develop hypertension. Pre-eclampsia and delivery before 34 weeks of gestation were associated with increased risk in both SGA categories, although confidence intervals overlapped with those of women not presenting with pre-eclampsia but who delivered before 34 weeks.

Recurrence risks of SGA were higher with delivery in the 1[st] pregnancy of infants with birthweights below the 5[th] percentile when compared to the recurrence risks associated with SGA infants between the 5[th] and the 10[th] percentile. Stillbirth in the 1[st] pregnancy was associated with a lower risk of SGA in the subsequent pregnancy in both categories (SGA below the 5[th] percentile adjusted OR 0.36, 95% CI 0.29–0.44; SGA between the 5[th] and 10[th] percentiles adjusted OR 0.57, 95% CI 0.47–0.69). The same occurred for neonatal mortality (SGA below the 5[th] percentile adjusted OR 0.74, 95% CI 0.59–0.93; SGA between the 5[th] and 10[th] percentiles adjusted OR 0.49, 95% CI 0.36–0.66).

**Table 2. Baseline characteristics at 1st pregnancy delivery by SGA.**

|  | Non-SGA | | SGA 5–10th percentile | | SGA <5th percentile | |
|---|---|---|---|---|---|---|
|  | (n = 239,447) | | (n = 11,991) | | (n = 13,593) | |
| Maternal age, years† | 29 | (26–31) | 29 | (26–31) | 29 | (25–31) |
| Gestational age at delivery, weeks† | 40 | (38–41) | 40 | (38–41) | 40 | (38–41)* |
| Non-caucasian, n (%) | 29,144 | 12.2% | 2,106 | 17.6% | 2,568 | 18.9% |
| Low socioeconomic status, n (%) | 61,951 | 25.9% | 2,722 | 22.7% | 2,965 | 21.8% |
| Hypertension, n (%) | 34,412 | 14.4% | 2,147 | 17.9% | 2,915 | 21.4% |
| Preeclampsia, n (%) | 5,041 | 2.1% | 584 | 4.9% | 745 | 5.5% |
| Chronic hypertension, n (%) | 2,318 | 1.0% | 160 | 1.3% | 201 | 1.5% |
| Chronic diabetes, n (%) | 2,530 | 1.1% | 53 | 0.4% | 69 | 0.5% |
| Gestational diabetes, n (%) | 1541 | 0.6% | 34 | 0.3% | 44 | 0.3% |
| Placental abruption, n (%) | 190 | 0.1% | 21 | 0.2% | 25 | 0.2% |
| HELLP syndrome, n (%) | 821 | 0.3% | 104 | 0.9% | 98 | 0.7% |
| Assisted reproduction, n (%) | 51,454 | 21.5% | 2,522 | 21.0%* | 3,001 | 22.1%* |
| Spontaneous labor, n (%) | 166,548 | 69.6% | 8,882 | 74.1% | 9,741 | 71.7% |
| Stillbirth, n (%) | 1,585 | 0.7% | 255 | 2.1% | 555 | 4.1% |
| Neonatal mortality, n (%) | 893 | 0.4% | 346 | 2.9% | 223 | 1.6% |
| Congenital abnormalities, n (%) | 5,389 | 2.3% | 405 | 3.4% | 753 | 5.5% |

SGA: small for gestational age. HELLP syndrome: hemolysis, elevated liver enzymes, and low platelet count syndrome

† Given as median and interquartile range.

* Not statistically different compared with non-SGA with a 95% confidence interval.

## Descriptive characteristics by delivery of an SGA infant in the first pregnancy

Table 2 presents baseline data according to 1st delivery of infants with birtweights higher than the 10th percentile versus delivery of SGA infants in the two analyzed ranges. Median maternal ages were similar to women who delivered as was the median gestational age at delivery. Non-Caucasian women, as well as women with socioeconomic status classified as higher than the 25th percentile were more likely to deliver an SGA infant. Hypertension, pre-eclampsia and chronic hypertension were associated with higher rates of SGA in the 1st pregnancy, while diabetes and gestational diabetes were associated with lower rates. HELLP syndrome and placental abruption occurred more frequently in association with SGA. Assisted reproduction rates were similar in the three groups. Stillbirth, neonatal mortality and congenital abnormalities were more common in the SGA groups as was spontaneous labor.

## Risk of late and early pre-eclampsia in thesecond pregnancy by hypertension and pre-eclampsia occurrence in the 1st pregnancy

Fig 2 shows the results of the multivariable regressions on the occurrence of late and early onset pre-eclampsia in the 2nd pregnancy by the presence of hypertension, pre-eclampsia, and delivery of an SGA infant in the first pregnancy. Women who did not present any of these risk factors had the lowest rate of pre-eclampsia occurrence in the 2nd pregnancy. Delivery of an SGA infant slightly increased the risk of late onset pre-eclampsia, although numbers remained small in absolute terms. Hypertension and pre-eclampsia in the 1st pregnancy were associated with large effect sizes for the ocurrence of pre-eclampsia in the 2nd pregnancy, although concurrent delivery of an SGA infant did not appear to impose additional risk given overlapping confidence intervals. The exception to this was delivery of an SGA infant with birthweight in

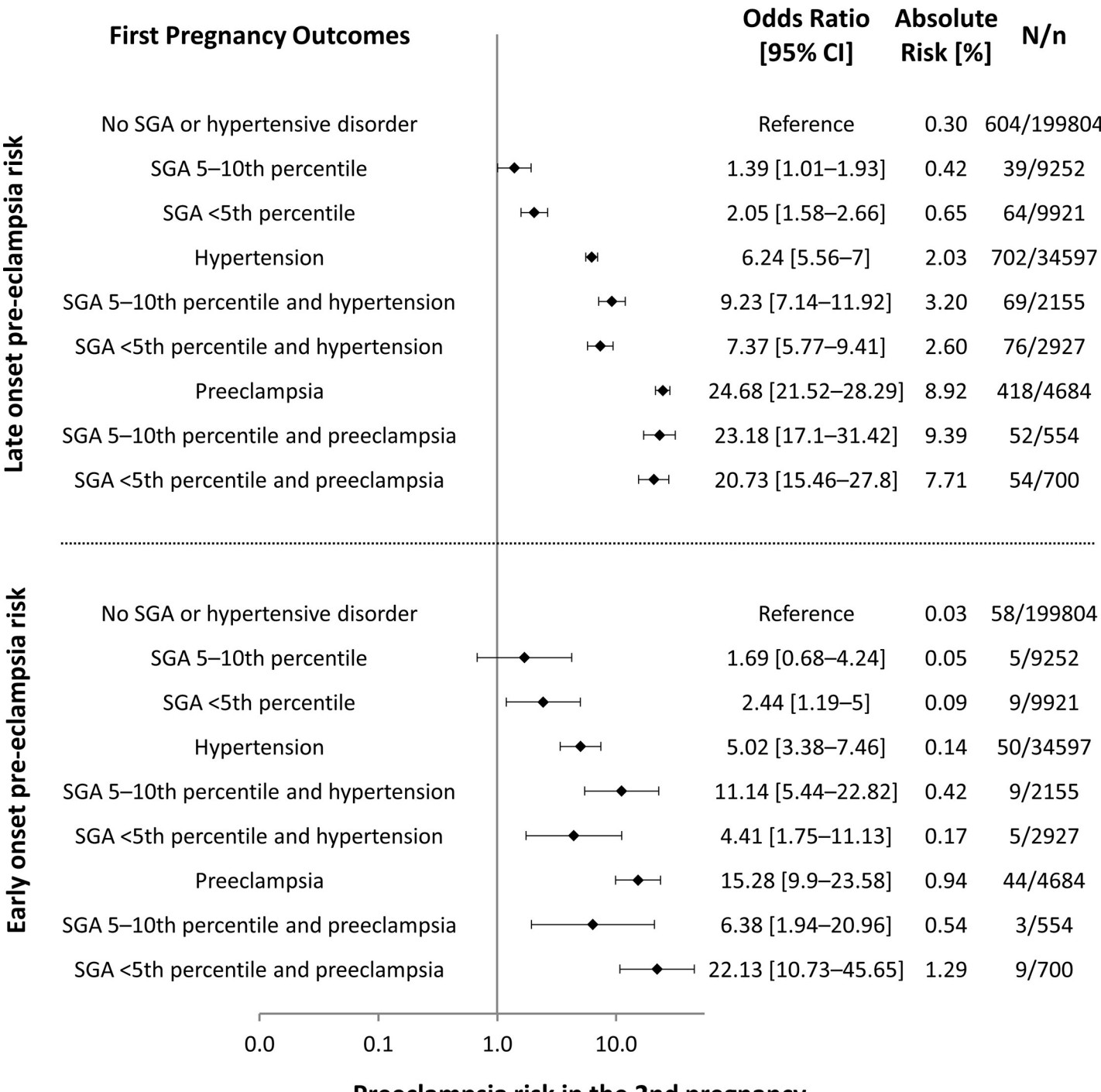

**Fig 2. Pre-eclampsia risk in the 2nd pregnancy by 1st pregnancy SGA, hypertension and pre-eclampsia.** Second pregnancy odds ratios and absolute risk of late onset pre-eclampsia (top) and early onset pre-eclampsia (bottom) by groups according to occurrence of small for gestational age, hypertension and pre-eclampsia in the first pregnancy. SGA: small for gestational age. CI: confidence interval.

the 5–10th in a pregnancy complicated by hypertension and the risk of 2nd pregnancy late onset pre-eclampsia, although taken in the context of the other results the likelihood of a false positive finding should be strongly considered.

Even in our large cohort, the occurrence of early onset pre-eclampsia in the 2nd pregnancy was a rare event. Second pregnancy delivery of SGA infants below the 5th percentile in the absense of 1st pregnancy hypertensive disorder was associated with increased risk, but absolute risks remained very small. Taking into account the less accurate point estimates due to the low number of events, the pattern of interaction between SGA, hypertension and pre-eclampsia was similar to that of late onset pre-eclampsia. We found no evidence of additional risk for early onset pre-eclampsia in the subsequent pregnancy due to delivery of an SGA infant if the first pregnancy was complicated by hypertension or pre-eclampsia.

## Discussion

### Main findings

Our results confirm, first of all, that the main risk factors for delivery of an SGA infant in the 2nd pregnancy is delivery of an SGA infant in the 1st pregnancy. For occurrence of pre-eclampsia in the 2nd pregnancy an SGA, the main risk factor is occurrence of pre-eclampsia in the 1st pregnancy. These established findings served as a basis for the comparisons of risks that this study focused on. [15–23]

In the present study, we found that in the absence of pre-eclampsia or hypertension, delivery of an SGA infant in the first pregnancy increased the risk of pre-eclampsia in the following pregnancy, although the absolute risk remained small. Women who developed pre-eclampsia and delivered an SGA infant in their first pregnancy had no higher risk of recurrence of pre-eclampsia than women who developed pre-eclampsia but delivered an infant with a birth-weight above the 10th percentile in their previous pregnancy. In other words, the strong risk of pre-eclampsia in the 2nd pregnancy imposed by its occurrence in the 1st pregnancy dominates the potential additional risk imposed by the delivery of an SGA infant in the 1st pregnancy. We have also shown that preterm delivery before the 34th week was associated with a higher risk of SGA in the subsequent pregnancy. We found no compelling evidence that delivery before the 34th week in the previous gestation further strongly compounded the risk of SGA if the woman also developed hypertension or pre-eclampsia in the 1st pregnancy. Although SGA risks are slightly higher in these situations, the overlapping confidence intervals and the small size effects remain unconvincing.

### Strengths and limitations

This study's main strength is that we used large sized cohort data, which was collected nation-wide and encompassed approximately 96% of all pregnancies and births that occurred within the analyzed period (2000–2007). The vast majority of Dutch perinatal caregivers contribute to Perined's data collection, with only 1–2% of general practitioners and 2–3% of midwives not reporting on pregnancies under their care. Nonetheless, this linked cohort dataset was found to accurately represent the Dutch national pregnancy and delivery outcomes.[31]

Because of the large size of the cohort, we were able to reliably evaluate the effects of hypertension, pre-eclampsia and early preterm delivery and the interaction between these risk factors for delivering an SGA infant in a subsequent pregnancy. We were also able to study the combination of rare events, such as recurrent pre-eclampsia and delivery of SGA infants. Thus, we provide further epidemiological evidence that could potentially serve to further clarify pathophysiological mechanisms that underlie the difference in timing of onset of pre-eclampsia and associated intrauterine growth restriction.

The use of SGA instead of intrauterine growth restriction is a common limitation found in the literature that is shared by our study. It is clear that one is an imprecise substitute for the other, as constitutionally small infants with no additional morbidity and mortality risks may

be wrongfully included in the population, while constitutionally large but growth restricted infants with a birthweight above a particular percentile may be excluded. We mitigated this problem by evaluating the efect of SGA delivery on the risk of pre-eclampsia in the subsequent pregnancy not only with the standard 10th percentile cut-off, but also with a cut-off at the 5th percentile. The 10th percentile allows easy comparison of the results between studies, while using the 5th percentile cut-off may be more rigorous with respect to identifying pathophysiological mechanisms, since it likely includes more births associated with truly pathological conditions and less constitutionally small infants. [22,23]

The effects of a number of potential confounders were taken into account, including those that are commonly excluded in other studies such as the presence of congenital anomalies, stillbirth and neonatal mortality. We considered the inclusion of these to be important since intrauterine growth restriction can be the result of multiple maternal and fetal issues, such as aneuploidies, congenital infections, and some placental and umbilical cord abnormalities, most of which are unlikely to play a significant role in pre-eclampsia risk in a subsequent pregnancy.[34–38] Furthermore, a priori exclusion of these three confounders would lead to misrepresentation of not only the cohort's SGA prevalences, but also of 2nd delivery pre-eclampsia occurrence. Perined records do not include or generally underreport additional confounders that would further enrich these analyses such as BMI, smoking, medication use, pre-existing vascular and kidney disease, history of thrombophilia, paternal influence and family history of PE.

The prevalence of pre-eclampsia in the 1st pregnancy in our data is likely to slightly underestimate the true prevalence in the Dutch population. This is because women who experienced pre-eclampsia in their 1st pregnancy and did not deliver a 2nd child within the data collection period were not included in our linked longitudinal dataset. The order of magnitude of this effect is uncertain, but data from a large Swedish cohort suggest that it may be small. The 1st pregnancy pre-eclampsia rate in that cohort decreased from 4.1% to 3.9% after exclusion of women who delivered only once.[39] As a final limitation, the identification of pre-eclampsia in our data was restricted by the absence of systolic blood pressure values in the analysed period. This likely caused further underestimation of pre-eclampsia in our study since isolated elevation of systolic blood pressure would be left out. However, this issue is compensated by Perined's independent recording of pre-eclampsia and eclampsia occurrences, which identifies women who satisfied the hypertension criterion for pre-eclampsia although diastolic blood pressure was in the normal range.

## Interpretation

A 2017 Cochrane review of 45 randomized controlled trials concluded that aspirin's potential as an effective intervention for the reduction of pre-eclampsia and intrauterine growth restriction is dependent on its early introduction. The authors found that low-dose aspirin had modest or no impact on pre-eclampsia and intrauterine growth restriction incidence when initiated after completion of 16 weeks of gestation.[40] This finding is supported by multiple previous studies and highlights the necessity of early identification of pregnant women at risk of developing either complication, and who consequently may benefit from introduction of aspirin before reaching this critical time limit.[41–43] The results of our study may help in the efforts to identify women that will benefit from the introduction of aspirin.

Bartsch et al. published in 2016 a study that combined data from large cohort studies in an attempt to systematically assess risk factors for pre-eclampsia that are easily identifiable before the 16th week.[18] Among the numerous risk factors evaluated, previous intrauterine growth restriction was the only one found to be not associated with increased risk of pre-eclampsia in

a succeeding pregnancy. This finding was based on a single Canadian cohort of 55,537 for whom history of prior IUGR was available. IUGR was defined in that study as birthweight below the 10[th] percentile according to the Canadian distribution plot. This method of assessment suffers from the same limitations present in our study discussed above, without considering effects for more severe SGA. Furthermore, of all women in the Canadian cohort, only 370 (0.7%) were identified to have this risk factor, whereas in our study the equivalent rate was 9.7%. This is likely one of the main reasons for the contrast with our findings.

Similar to our study, Voskamp et al. studied the recurrence of SGA using Dutch registry data. The authors concluded that women with hypertensive disorders in the 1[st] pregnancy and women who delivered an SGA infant in the 1[st] pregnancy were both at increased risk of SGA in the following pregnancy. Our results concur with the latter association, as do other studies, but the association regarding hypertensive disorders should be more nuanced.[20–22] As we were able to evaluate the impact of the previous gestational age at delivery, type of hypertensive disorder present and the interaction between these two factors, we were able to show that, other than history of SGA delivery, the main risk factor for SGA in a subsequent pregnancy is early preterm delivery, i.e., delivery before the 34[th] week of gestation. After adjustment for these two factors, their interaction, and numerous other risk factors, the presence of hypertension in the 1[st] pregnancy was not associated with increased risk of SGA in the subsequent pregnancy, unless in association with early preterm delivery.

## Conclusion

Our finding that SGA delivery in a previous pregnancy is associated with increased risk of early and late onset pre-eclampsia even in the absense of hypertension and pre-eclampsia adds credibility to the hypothesis of common pathological mechanisms. Evidence linking early onset pre-eclampsia to increased risk of SGA in a subsequent pregnancy is more limited, since we found that women who delivered preterm without hypertensive disorders had similar increased risks. Nonetheless, it is clear that women who previously presented these complications may benefit from the introduction of low-dose aspirin before the 16[th] week of gestation for the prevention of pre-eclampsia and SGA.

## Author Contributions

**Conceptualization:** Thomas P. Bernardes, Ben W. Mol, Paul van den Berg, Henk Groen.

**Data curation:** Thomas P. Bernardes, Anita C. J. Ravelli.

**Formal analysis:** Thomas P. Bernardes, Henk Groen.

**Methodology:** Thomas P. Bernardes, Anita C. J. Ravelli, Henk Groen.

**Resources:** Ben W. Mol, Paul van den Berg, H. Marike Boezen, Henk Groen.

**Supervision:** Ben W. Mol, Paul van den Berg, H. Marike Boezen, Henk Groen.

**Writing – original draft:** Thomas P. Bernardes, Henk Groen.

**Writing – review & editing:** Thomas P. Bernardes, Ben W. Mol, Anita C. J. Ravelli, Paul van den Berg, H. Marike Boezen, Henk Groen.

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
