## [Decision Letter · Decision Letter 0]

2 Sep 2019

PONE-D-19-18526

Early and late onset pre-eclampsia and small for gestational age risk in subsequent pregnancies.

PLOS ONE

Dear Authors,

Thank you for submitting your manuscript to PLOS ONE. After careful consideration, we feel that it has merit but does not fully meet PLOS ONE’s publication criteria as it currently stands. Therefore, we invite you to submit a revised version of the manuscript that addresses the points raised during the review process.

We would appreciate receiving your revised manuscript by 1st October. To enhance the reproducibility of your results, we recommend that if applicable you deposit your laboratory protocols in protocols.io, where a protocol can be assigned its own identifier (DOI) such that it can be cited independently in the future. For instructions see: http://journals.plos.org/plosone/s/submission-guidelines#loc-laboratory-protocols

We look forward to receiving your revised manuscript.

Kind regards,

Salvatore Andrea Mastrolia, M.D.

Academic Editor

PLOS ONE

1. We note that you have indicated that data from this study are available upon request. PLOS only allows data to be available upon request if there are legal or ethical restrictions on sharing data publicly. For information on unacceptable data access restrictions, please see http://journals.plos.org/plosone/s/data-availability#loc-unacceptable-data-access-restrictions.

Reviewers' comments:

Reviewer's Responses to Questions

**Comments to the Author**

1. Is the manuscript technically sound, and do the data support the conclusions?

Reviewer #1: Yes

Reviewer #2: Yes

2. Has the statistical analysis been performed appropriately and rigorously? 

Reviewer #1: No

Reviewer #2: Yes

3. Have the authors made all data underlying the findings in their manuscript fully available?

Reviewer #1: Yes

Reviewer #2: Yes

4. Is the manuscript presented in an intelligible fashion and written in standard English?

Reviewer #1: Yes

Reviewer #2: Yes

5. Review Comments to the Author

Reviewer #1: Early and late onset preeclampsia and small for gestational age risk in subsequent pregnancy

The study aims to evaluate whether delivery of an SGA infant affected the risk of early and late onset pre-eclampsia in a subsequent pregnancy, and conversely, if occurrence of early and late onset pre-eclampsia in the previous pregnancy increases SGA risk.

Overall, this study touches upon an important topic and, together with previous studies describes the association between a history of preeclampsia and subsequent delivery of an SGA infant.

However, this topic has been extensively described in the North American and European literature and there is no reason to suspect different outcomes in same population.

The following problems are most troubling when considering this article for publication:

1. Alternating the care between the Dutch perinatal caregivers, general practitioners and midwives cannot be ruled out. Moreover, is it possible that the higher risk population's care will be provided by the perinatal caregivers as opposed to midwifery care, for example?

2. Data regarding other related factors such as BMI and other important risk factors for preeclampsia such as vascular, kidney of autoimmune disease was not provided and not controlled for.

3. Pre-gestational diabetes is a potential confounder and an important risk factor for preeclampsia and was not controlled for in the multivariable logistic regression model.

4. No description of those lost to follow up between consecutive pregnancies was provided.

The importance of this manuscript to the medical literature-

Many studies have investigated and described the common pathophysiological features that favor parallel occurrence of preeclampsia and growth restriction. Moreover, it is well established that women with pre-eclampsia or SGA in a previous pregnancy have high risk of recurrence. Therefore, one should consider whether this study presents an important addition to the medical literature that has faced many studies dealing with the same topic in different populations in the world.

Reviewer #2: Well written and methodologically correct manuscript on association between pre-eclampsia and SGA.

In figure 1, the increased risk of SGA (both <p5 as="" p5-10="">p10 in the first pregnancy? Or is the increased risk of SGA in the subsequent pregnancy merely the result of recurrence of SGA in these groups? This should be explained more clearly or added in the analysis.

Hereby, it should also be taken into account that birthweight charts are influenced by the association between placental pathology and preterm delivery. Consequently the amount of SGA infants tends to be underestimated in the preterm period.

Also I suggest that for clinical use, some findings should/could be highlighted more clearly.

For both patients and clinicians absolute risks are more useful than relative risks or odds ratios.

For example: the odds ratios for both early onset and late onset pre-eclampsia in the second pregnancy in women with a previous pregnancy complicated by SGA <p5 and="" pre-eclampsia="">  This could be done with a flow chart that can be used in national guidelines and local protocols. (first pregnancy hypertension Yes/no  first pregnancy pre-eclampsia yes/no  first pregnancy SGA <p5 --="" no="" yes=""> and subsequent risks of 2nd pregnancy Pre-eclampsia and SGA.

Other comments/sugggestions:

- Abstract:

conclusion "Women with 1st pregnancy early onset pre-eclampsia have increased risk of SGA <5th percentile

45 in the 2nd pregnancy." is not complete/correct. This should be something like: "Women with 1st pregnancy early onset hypertension, pre-eclampsia or delivery <34 weeks have increased risk of SGA (<5th and 5th-10th percentile) in the 2nd pregnancy."

conclusion: "SGA in the 1st pregnancy increases pre-eclampsia risk in the 2nd pregnancy even in the absence of hypertensive disorders in the 1st pregnancy" should in my oppinion be adjusted: "SGA in the 1st pregnancy increases pre-eclampsia risk in the 2nd pregnancy even in the absence of hypertensive disorders in the 1st pregnancy the is an increased risk of pre-eclampsia in the 2nd pregnancy if the first born had a birthweight below the 5th percentile... however, absolute risks are very low (<0.1% for early onset pre-eclampsia, and <0.7% for late onset pre-eclampsia)"

- Introduction

Line 51: "The severity of adverse outcomes has strong association..." ==> consider changing to: "The severity of adverse outcomes has a strong association..."

 </p5></p5></p5>

6. PLOS authors have the option to publish the peer review history of their article (what does this mean?). If published, this will include your full peer review and any attached files.

Reviewer #1: No

Reviewer #2: No

---

## [Author Response · Author response to Decision Letter 0]

6 Feb 2020

Reviewer #1

Q1. Alternating the care between the Dutch perinatal caregivers, general practitioners and midwives cannot be ruled out. Moreover, is it possible that the higher risk population's care will be provided by the perinatal caregivers as opposed to midwifery care, for example?

Answer: Yes, as described in the first paragraph of the Methods section, the Perined national Dutch registry is composed of linked data from three different registries: the midwifery registry (LVR1), the obstetrics registry (LVR2), and the neonatology registry (LNR). As a result, it encompasses 96% of all deliveries that occur in the Netherlands.

In a population-based cohort study it is to be expected that care is provided through all levels of complexity and risk as represented in the health system under study. This study is no exception and reflects the particularities of obstetric care in the Dutch system. Primary care is offered by midwives and general practitioners, who refer high risk women to secondary and tertiary care, representing general and academic hospitals, respectively. Overall, we feel that our data are a good reflection of the case-mix and care system for pregnant women in The Netherlands.

Q2. Data regarding other related factors such as BMI and other important risk factors for preeclampsia such as vascular, kidney of autoimmune disease was not provided and not controlled for.

Answer: As mentioned in our Discussion section under the “Strengths and limitations” sub-heading: 

“Perined records do not include or generally underreport additional confounders that would further enrich these analyses such as BMI, smoking, medication use, pre-existing vascular and kidney disease, history of thrombophilia, paternal influence and family history of PE”. 

We agree that access to this information would be helpful for the analyses, but they are unfortunately not available in the population-based dataset.

3. Pre-gestational diabetes is a potential confounder and an important risk factor for preeclampsia and was not controlled for in the multivariable logistic regression model.

Answer: Pre-gestational diabetes was included as a potential confounder and controlled for in the multivariable logistic regression models as stated in our Methods section. We have noticed that in the same section we did not mention pre-gestational diabetes as a variable that we have controlled for in that section. Consequently, we have added this information to the relevant sentence and it now reads as follows:

To adjust for potential confounders the following 1st pregnancy clinical and demographic characteristics were included in the regressions: maternal age (years), gestational age at delivery (weeks and before/after 34 weeks of gestational age), non-Caucasian ethnicity (yes or no), low socioeconomic status (yes or no), any cause hypertension (yes or no), pre-eclampsia (yes or no), chronic hypertension (yes or no), pre-gestational diabetes (yes or no), gestational diabetes (yes or no), placental abruption (yes or no), HELLP syndrome (yes or no), assisted reproduction (yes or no), spontaneous labor (yes or no), stillbirth (yes or no), neonatal mortality (yes or no), congenital abnormalities (yes or no).

Q4. No description of those lost to follow up between consecutive pregnancies was provided.

Answer: By construction, as a result of the probabilistic record linkage procedure used to match 1st and 2nd deliveries, there is no loss to follow up for the women included in the linked dataset. If the calculated posterior probability of a proper match between 1st and 2nd deliveries was higher than 80% it was considered high enough to assume it belonged to the same mother. Full description of the methods used in this procedure are referenced in the manuscript and found here: 

Schaaf JM, Hof MHP, Mol BWJ, Abu-Hanna A, Ravelli ACJ. Recurrence risk of preterm birth in subsequent singleton pregnancy after preterm twin delivery. Am J Obstet Gynecol. 2012 Oct;207(4):279.e1-7.

Reviewer #2

Q1. In figure 1, the increased risk of SGA (both p10 in the first pregnancy? Or is the increased risk of SGA in the subsequent pregnancy merely the result of recurrence of SGA in these groups? This should be explained more clearly or added in the analysis.

Hereby, it should also be taken into account that birthweight charts are influenced by the association between placental pathology and preterm delivery. Consequently the amount of SGA infants tends to be underestimated in the preterm period.

Answer: Results presented in figure 1 correspond to the multivariable logistic regression models that evaluate 2nd delivery risk of SGA infants based on data of the 1st pregnancy. In particular, we were interested in assessing the effects of pre-eclampsia, hypertension and gestational age at delivery on this risk. To more accurately find the effect-sizes of each particular factor, we included in the model variables such as delivery of an SGA infant in the 1st pregnancy, a well-established risk factor for subsequent SGA delivery.

The advantage of using a multivariable model in the evaluation of multifactorial issues such as the delivery of an SGA infant is that we can control for the influence of factors we have data available for. In this sense, the increased risk of SGA in the 2nd pregnancy associated with SGA delivery in the 1st pregnancy found in figure 1 represents the risk associated with this factor controlled for pre-eclampsia, hypertension, delivery before or after 34 weeks and all the other factors listed in our Methods section such as maternal age, diabetes (chronic and gestational) and congenital abnormalities. This follows for all other variables in the model.

The influence of pathology and preterm delivery on birthweight is the raison d'être for the construction of birthweight charts and definitions such as SGA. With this in mind, the inclusion of variables such as congenital abnormalities and placental abruption as variables in our models no doubt adds strength to the results.

Q2. Also I suggest that for clinical use, some findings should/could be highlighted more clearly. For both patients and clinicians absolute risks are more useful than relative risks or odds ratios.

Answer: We agree to an extent and that is why we have provided absolute risks in Figures 1 and 2. We believe though that these should be interpreted in the wider context of a multifactorial problem and that requires the inclusion of estimates such as odds ratios adjusted in a multivariable model.

Q3. "Women with 1st pregnancy early onset pre-eclampsia have increased risk of SGA <5th percentile

45 in the 2nd pregnancy." is not complete/correct. This should be something like: "Women with 1st pregnancy early onset hypertension, pre-eclampsia or delivery <34 weeks have increased risk of SGA (<5th and 5th-10th percentile) in the 2nd pregnancy."

Answer: Although we appreciate the suggestion, we prefer to limit our abstract conclusion to refer specifically to the findings that pertain to our original objectives. These were defined in the abstract as: 

To investigate whether delivery of a small for gestational age (SGA) infant in the 1st pregnancy increases the risk of early and late onset pre-eclampsia in the 2nd pregnancy. Conversely, we investigated whether pre-eclampsia in the 1st pregnancy impacts SGA risk in the 2nd pregnancy.

Q4. conclusion: "SGA in the 1st pregnancy increases pre-eclampsia risk in the 2nd pregnancy even in the absence of hypertensive disorders in the 1st pregnancy" should in my opinion be adjusted: "SGA in the 1st pregnancy increases pre-eclampsia risk in the 2nd pregnancy even in the absence of hypertensive disorders in the 1st pregnancy the is an increased risk of pre-eclampsia in the 2nd pregnancy if the first born had a birthweight below the 5th percentile... however, absolute risks are very low (<0.1% for early onset pre-eclampsia, and <0.7% for late onset pre-eclampsia)"

Answer: We appreciate the suggestion and have adjusted the abstract conclusion as follows:

SGA in the 1st pregnancy increases pre-eclampsia risk in the 2nd pregnancy even in the absence of hypertensive disorders in the 1st pregnancy, although absolute risks remain low.

Q5.Introduction

Line 51: "The severity of adverse outcomes has strong association..." ==> consider changing to: "The severity of adverse outcomes has a strong association..."

Answer: We appreciate the suggestion and have adjusted the introduction accordingly.

---

## [Decision Letter · Decision Letter 1]

3 Mar 2020

Early and late onset pre-eclampsia and small for gestational age risk in subsequent pregnancies.

PONE-D-19-18526R1

Dear Authors,

We are pleased to inform you that your manuscript has been judged scientifically suitable for publication and will be formally accepted for publication once it complies with all outstanding technical requirements.

With kind regards,

Salvatore Andrea Mastrolia, M.D.

Academic Editor

PLOS ONE

Reviewers' comments:

Reviewer's Responses to Questions

**Comments to the Author**

1. If the authors have adequately addressed your comments raised in a previous round of review and you feel that this manuscript is now acceptable for publication, you may indicate that here to bypass the “Comments to the Author” section, enter your conflict of interest statement in the “Confidential to Editor” section, and submit your "Accept" recommendation.

Reviewer #1: All comments have been addressed

Reviewer #2: All comments have been addressed

2. Is the manuscript technically sound, and do the data support the conclusions?

Reviewer #1: Yes

Reviewer #2: Yes

3. Has the statistical analysis been performed appropriately and rigorously? 

Reviewer #1: Yes

Reviewer #2: Yes

4. Have the authors made all data underlying the findings in their manuscript fully available?

Reviewer #1: Yes

Reviewer #2: Yes

5. Is the manuscript presented in an intelligible fashion and written in standard English?

Reviewer #1: Yes

Reviewer #2: Yes

6. Review Comments to the Author

Reviewer #1: (No Response)

Reviewer #2: All comments have been addressed. In my opinion the manuscript is methodologically sound and well written.

However some point need attention or adjustment:

Figures:

figure1 and figure2 should be adjusted in such a way that it is more obvious which factors concern the first pregnancy, and which concern the second pregnancy. Especially the vertical "Late onset pre-eclampsia" and "early onset pre-eclampsia" on the left in the figure are confusing ==>

The figure could be separated into.

Figure 1a => "influence of first pregnancy outcomes on late onset pre-eclampsia risk in the second pregnancy"

Figure 1b => "influence of first pregnancy outcomes on early onset pre-eclampsia risk in the second pregnancy"

The same applies to figure 2.

Conclusion:

"SGA in the 1st pregnancy increases pre-eclampsia risk in the 2nd pregnancy even in the absence of hypertensive disorders in the 1st pregnancy, although absolute risks remain low."

Although statistically siginificant. The clinical relevance of this finding remains very doubtful (absolute risk of 0.09% and 0.64%). The finding could also merely be a result of stricter monitoring of pregnant women who delivered a (unexpected) SGA baby in their first pregnancy.

Example: Mother with Diastolic BP of 88mmHg and proteinuria (that wasn't tested because BP was <90 and no symptoms) delivers an SGA baby in her first pregnancy. She is monitored more strictly in her second pregnancy and proteinuria is diagnosed and diastolic BP measured at 90mmHg. This should be addressed in the discussion and the conclusion should be adjusted, either by removing this complete sentence, or by adding absolute risks.

7. PLOS authors have the option to publish the peer review history of their article (what does this mean?). If published, this will include your full peer review and any attached files.

Reviewer #1: Yes: Pariente G, Department of Obstetrics and Gynecology, Soroka University Medical Center, Beer - Sheva, Israel

Reviewer #2: No

---

## [Editor Report · Acceptance letter]

16 Mar 2020

PONE-D-19-18526R1 

Early and late onset pre-eclampsia and small for gestational age risk in subsequent pregnancies. 

Dear Dr. Bernardes:

I am pleased to inform you that your manuscript has been deemed suitable for publication in PLOS ONE. Congratulations! Your manuscript is now with our production department. 

With kind regards,

on behalf of

Dr. Salvatore Andrea Mastrolia 

Academic Editor

PLOS ONE